



# Spatiotemporal variation and trends of equivalent black carbon in the Helsinki metropolitan area in Finland

Krista Luoma[1], Jarkko V. Niemi[2], Aku Helin[3], Minna Aurela[3], Hilkka Timonen[3], Aki Virkkula[3], Topi Rönkkö[4], Anu Kousa[2], Pak Lun Fung[1], Tareq Hussein[1,5] and Tuukka Petäjä[1]

[1]Institute for Atmospheric and Earth System Research / Physics, Faculty of Science, University of Helsinki, P.O. Box 68, 00014 Helsinki, Finland
[2]Helsinki Region Environmental Services Authority, P.O. Box 100, 00066 Helsinki, Finland
[3]Atmospheric Composition Research, Finnish Meteorological Institute, P.O. Box 503, 00101 Helsinki, Finland
[4]Aerosol Physics Laboratory, Faculty of Engineering and Natural Sciences, Tampere University, P.O. Box 692, 33014 Tampere, Finland
[5]Department of Physics, The University of Jordan, 11942 Amman, Jordan

*Correspondence to*: Krista Luoma (krista.q.luoma@helsinki.fi)

**Abstract:** In this study, we present results of 12 years of black carbon (BC) measurements at 14 different measurement sites around the Helsinki metropolitan area (HMA) and at one background site outside the HMA. The main local sources of BC in the HMA are traffic, and residential wood combustion in fireplaces and sauna stoves. All the BC measurements were conducted optically and therefore we refer to the measured BC as equivalent BC (eBC). Measurement stations were located at different types of environments that represented traffic environment (six sites), detached housing area (five sites), urban background (two sites), and regional background (two sites). The measurements of eBC were conducted during 2007 – 2018; however, the time period and the length of the time series varied from site to site. As expected, the largest annual mean eBC concentrations were measured at the traffic sites (0.67 – 2.64 µg m$^{-3}$) and the lowest at the regional background sites (0.16 – 0.29 µg m$^{-3}$). The annual mean eBC concentrations at the detached housing sites varied in the range of 0.64 – 0.80 µg m$^{-3}$ and the annual mean eBC concentrations at the urban background sites varied in the range of 0.42 – 0.68 µg m$^{-3}$. The clearest seasonal variation was observed at the detached housing sites, where the residential wood combustion increased the eBC concentrations during the cold season. Traffic rates and wood burning influenced the diurnal and weekly variations of eBC concentration in different types of environments. The dependency was not so clear for the other air pollutants, which were here NO$_x$ and mass of particles smaller than 2.5 µm (PM$_{2.5}$). At four sites, which had at least four-year-long time series available, we observed that the eBC concentrations had statistically significant decreasing trends, which varied in the range of -10.4 – -5.9 % yr$^{-1}$. Compared to the trends determined at the urban and regional background sites, the absolute trends decreased the fastest at the traffic sites and especially during the morning rush hour. The relative long-term trends of eBC and NO$_x$ were similar to each other, and their concentrations decreased more rapidly than the concentration of PM$_{2.5}$. The results indicate that especially the emissions from traffic have decreased in the HMA during the last decade. This shows that air pollution control, new emission standards and newer fleet of vehicles really have an effect in the air quality.



## 1 Introduction

Air pollution is one of the biggest environmental health risks in the world. Air pollution consists of both gaseous components and particulate matter (PM). Lelieveld et al. (2015) estimated that particles smaller than 2.5 µm in diameter ($PM_{2.5}$) and ozone ($O_3$) together caused about 3.3 million premature deaths globally in 2010. Majority of these premature deaths were due to

$PM_{2.5}$ (approximately 1.9 million). More than 65 % of the premature deaths caused by $PM_{2.5}$ were related to cardiovascular diseases, the second largest cause for the $PM_{2.5}$-related premature deaths were lung and respiratory diseases, and a small fraction of the premature deaths were due to lung cancer (Lelieveld et al., 2015). The causes for the adverse health effects of PM are the PM induced inflammation and toxic materials, which are transported in the respiratory system by the particles.

WHO reported that PM emitted from combustion sources, especially from traffic, is more harmful for health than PM from other sources (Krzyżanowski et al., 2005). Typical combustion sources, which are also discussed in this study, are traffic and domestic wood burning. Traffic emits a complex mixture of gaseous and fine particulate compounds (Rönkkö and Timonen, 2019). Domestic wood burning also emits fine particles including toxic compounds, such as benzo($a$)pyrene (Hellén et al., 2017). Black carbon (BC), which is defined as black carbonaceous particulate matter, is a good indicator of pollution from

combustion because it is a side product of incomplete combustion. Therefore, it is useful to measure BC alongside with the other air quality parameters since BC concentration can give additional information about the health effects of aerosol particles than just PM, which does not only originate from combustion sources (Janssen et al., 2011). BC is not just an indicator on bad air quality, but the BC particles themselves also have adverse health effects. BC particles, which are around the size range of ~100 nm, can penetrate deep in the respiratory system and all the way in the alveolar region, from where the particles can be

transported in the blood circulation system and further on in the organs.

Due to its black appearance, BC absorbs solar radiation and decreases the albedo of reflecting surfaces (i.e., snow and ice sheets). BC has been estimated to be one of the largest warming agents in the climate change (Stocker et al., 2013). Since BC is emitted in the air as particles, its lifetime in the atmosphere is relatively short (few weeks) compared to greenhouse gases

(tens of years). Therefore, in addition to improving air quality, cutting down the BC emissions would have rather fast effects in the radiative forcing slowing down the rate of global warming. In order to reduce BC emissions there has been, for example, a suggestion of implementing a BC footprint similar to the $CO_2$ footprint (Timonen et al., 2019).

Several recent studies have reported decreasing long-term trends of BC concentration in different types of environments

including urban areas (e.g., Kutzner et al., 2018; Singh et al., 2018). In Finland, the concentration of BC and its trends has been studied especially in background sites (Hienola et al., 2013; Hyvärinen et al., 2011), but not that much in urban areas. The previous studies about BC concentration in the Helsinki Metropolitan Area (HMA) are mainly from shorter campaigns





(Aurela et al., 2015; Dos Santos-Juusela et al., 2013; Helin et al., 2018; Järvi et al., 2008; Pakkanen et al., 2000; Pirjola et al., 2017).

The objective of this study is to investigate the spatio-temporal variability of BC in different types of environments in the HMA. Our objective is also to clarify how the proximity of combustion sources, which in HMA are mainly traffic and domestic wood burning (Helin et al., 2018; Savolahti et al., 2016), affect the BC concentrations. In this study, we utilize BC data measured at various different locations in the HMA and at one site outside the HMA. The measurements were conducted during 2007 – 2018. To study how the variability of BC differs from the variability of the monitored and regulated air quality parameters, we also included parallel measurements of $PM_{2.5}$ and $NO_x$ in this study. Measuring BC in addition to $PM_{2.5}$ gives a better view on the effect of local pollution sources and more hazardous air pollutants. Understanding the anthropogenic sources of pollution helps overcoming the problems related to poor air quality.

## 2 Measurements and methods

### 2.1 The field sites

The HMA consists of four different cities (Helsinki, Espoo, Vantaa, and Kauniainen) and it is the most densely populated area in Finland. The total population in the HMA is about 1.2 million. Helsinki is the capital of Finland and it is located in the southern coast of the country (60°10'N, 24°56'E). The HMA locates on the seaside of the Gulf of Finland and therefore the climate in the area is a transition zone between oceanic and continental climate and typically defined as humic continental climate. The coldest monthly mean temperature (-5 °C) during the measurement period (2009 – 2018) occurred in February and the warmest monthly mean temperature (19 °C) in July. The mean wind speed over the whole measurement period was about 4 m s$^{-1}$. For more detailed information about the meteorological parameters, see Fig. S1.

The measurements of BC concentration were conducted at 15 sites. We classified the stations into four categories: traffic environment (TR), detached housing area (DH), urban background (UB), and regional background (RB). The locations of the measurement sites are presented in Figure 1; the coordinates and aerial photos of the stations are provided in Sect. S2.

One of the sites, which represented the RB, was located circa 200 km north of the HMA in Hyytiälä. At some of the stations, the measurements have been repeated or conducted on a long-term basis and at some of the stations, the measurements lasted only for one year. The measurement periods for each site are shown in Table 1. Most of the stations were operated by the Helsinki Region Environmental Services Authority (HSY), which is the authority monitoring the air quality in the HMA.



### 2.1.1 Traffic stations

Six stations (TR1 – TR6), located close to a busy street or road (Fig. S3), were categorized as traffic stations. At these sites, traffic was the dominating source of pollution. Detailed information, such as traffic counts (TC), fraction of heavy-duty (HD; i.e., trucks and buses), and the distance of the station to the closest traffic lines are presented in Table 2.

5  TR1 located right in the city center and the station represented the concentration level that pedestrians are exposed to in the city center. TR2 and TR3 located in street canyons, where tall buildings frame the streets. TR2 was located next to Mäkelänkatu, which is one of the main streets leading to Helsinki city center, and TR3 was located in the northern part of Mannerheimintie, another main street to Helsinki city center. In addition to passenger cars, many busses (heavy-duty) bypass the TR2 and TR3.

While, TR1, TR2, and TR3 were in or close to the Helsinki city center, TR4 was located next to a ring road, Kehä I, which is a highway going around the Helsinki city. The station represented the concentration level on walkways and bus stops that are next to major roads.

15  TR5 and TR6 were located close to busy streets and intersections in the city centers of Vantaa and Espoo, respectively. These city centers were not as large as the city center of Helsinki, but there were still busy roads and apartment buildings. As seen in Fig. 1, TR5 and TR6 were closer to the detached housing areas than the other TR stations.

### 2.1.2 Stations in the detached housing areas

20  Stations DH1, DH2, DH3, DH4, and DH5 were located in Vartiokylä, Ruskeasanta, Lintuvaara, Rekola, and Itä-Hakkila, respectively (Fig. S4). These residential areas consisted of separate one-family houses, small streets, and some forests and parks. The traffic rates at the small streets next to these stations were low. For example, the traffic rates on the streets next to DH1 and DH5 were estimated to be about 2 600 vehicles per day, which is notably less than close to the TR sites (Table 2). In 2014, the HSY conducted a large survey about wood combustion in the detached houses in the HMA (HSY, 2016, in Finnish only; Hellén et al., 2017). In the study, HSY estimated that 90 % of the households burn wood to warm up houses and/or saunas. However, less than 2 % of the households use wood burning as the main heating source.

### 2.1.3 Urban background stations

The UB stations represented the concentration levels that people are generally exposed to in urban areas in Helsinki. The UB 30  stations, UB1 and UB2, were located in the close to the central area but not in vicinity of busy roads (Fig. S5a-b). UB1 was located in the Kallio district next to a sports field and close to Helsinginkatu, a moderately busy street, where the average



working day traffic count was about 5 000 vehicles per day (5 % heavy-duty). The area consisted mostly of apartment buildings. UB2 was located in Kumpula, in the campus of the University of Helsinki. UB2 is also known as SMEAR III (Station for Measuring Ecosystem-Atmosphere Relations; Järvi et al., 2009) and it is run in collaboration by the University of Helsinki and Finnish Meteorological Institute (FMI). There was a busy street, Hämeentie, located after 200 m wide forest

band. The traffic count at Hämeentie was about 40 000 vehicles per day.

### 2.1.4 Regional background stations

The RB stations represented the concentration levels outside the urban area without any main local sources (Fig. S5c-d). The RB stations were mainly affected by regional and long-range transported pollutants. RB1 was located in Luukki, which is a

rural area in Espoo about 23 km away from the Helsinki city center. RB2 was located almost 200 km away from Helsinki in Hyytiälä. RB2 was stationed in a boreal forest, far away from any pollution sources. RB2 was included in this study to compare the pollution levels in a city to those observed in a remote countryside station. RB2 is part of the SMEAR network and it is also known as SMEAR II (Hari and Kulmala, 2005).

### 2.1.4 Meteorological station

The meteorological station measuring wind direction (WD), wind speed (WS), temperature (T), pressure (p), relative humidity (RH), and precipitation, was located in a rooftop (78 m a.s.l) in the Pasila district close to central Helsinki. In this study, we used the measurements of meteorological parameters conducted in Pasila to represent the meteorological conditions of all the stations located in the HMA. Also, the air quality stations measured the meteorological parameters, but four meters above the

ground where the local factors, such as trees and buildings, may affect the measurements.

### 2.2 Measurements of equivalent black carbon

HSY arranged the measurements at 13 of the sites, University of Helsinki conducted the measurements at RB2, and the Finnish Meteorological Institute (FMI) together with the University of Helsinki conducted the measurements at UB2. All the BC

measurements were conducted optically and the BC concentration was derived from the light absorption of the particles and hence we refer to the measured BC as equivalent black carbon (eBC; Petzold et al., 2013). At all of the measurement stations, the head of the sampling line was located 4 m above the ground. The concentration of eBC was measured for particles smaller than 1 µm ($PM_1$). However, at DH1 the eBC concentration was measured for $PM_{2.5}$ for first half of the year, but since most of the eBC mass concentration falls in the $PM_1$ (Vallius et al., 2000), the cut-off size should not cause big difference in the results.

Sample air was dried with an external dryer or by warming up the sample to 40 °C at most of the stations, but at TR1, UB2, DH5 and DH4 (only the first half of year) the sample air was not dried. Even if there was no drier, the sample air warmed up



to the room temperature, which decreases the RH, when the outdoor temperature is lower than the indoor temperature (i.e., the sample air is dried passively during the cold period, however, in summer, when the temperature difference is smaller, the RH does not necessarily decrease). At 11 of the stations, the measurements of eBC were conducted by using only a Multi-angle absorption photometer (MAAP; Thermo Fisher Scientific, model 5012), and at four of the stations (DH4, DH5, and RB2) all

or at least part of the measurements were conducted by using an Aethalometer (Magee Scientific, models AE31 and AE33). The instruments used at different sites are listed in Table S1.

The MAAP determines the absorption coefficient of aerosol particles by collecting the particles on a filter medium and by measuring the intensity of light penetrating the filter and the intensity of light that is scattered from the filter in two different

angles (Petzold and Schönlinner, 2004). The absorption coefficient is determined from these measurements by using a radiative transfer scheme. The eBC concentration is obtained from the absorption coefficient by using a mass absorption cross-section (MAC) value of 6.60 $m^2\,g^{-1}$ (Petzold and Schönlinner, 2004).

The Aethalometer measures the eBC concentration at seven wavelengths (370, 470, 520, 590, 660, 880, and 950 nm) and in

this study the wavelength 880 nm was used. The Aethalometer collects the sample aerosol particles on the filter material similarly to the MAAP, but unlike the MAAP, the Aethalometer only measures the attenuation of light through the filter. Therefore, the Aethalometer is prone to error caused by the increasing filter loading. In the newer model, AE33, this is automatically taken into account in real time as the instrument applies the so-called "dual-spot correction" to the data (Drinovec et al., 2015). For AE33, MAC value of 7.77 $m^2\,g^{-1}$ at 880 nm is recommended (Drinovec et al., 2015). For the older model,

AE31, a correction algorithm needs to be applied by the user (e.g., Collaud Coen et al., 2010). AE31 determines the BC concentration from the so-called attenuation coefficient and it uses mass attenuation cross-section value of 16.62 $m^2\,g^{-1}$ at 880 nm.

At DH4, model AE33 Aethalometer was used for the first half of the measurement period (1 January 2017 – 5 May 2017) and

at DH5, the whole data set was measured with an AE33. At these two stations, HSY corrected the eBC concentration by multiplying the concentration by 0.75, according to a comparison with MAAP. At RB2, an older model AE31 was used. The AE31 was first corrected for the filter loading error by using the correction algorithm suggested by Virkkula et al. (2007). After the filter loading correction, a comparison with MAAP showed that the AE31 data had to be multiplied by 1.08 to acquire similar concentrations (Sect. S3).

## 2.3 Measurements of NO$_x$ and PM$_{2.5}$

NO$_2$ and the mass of particles smaller than 2.5 µm (PM$_{2.5}$) are regulated pollutants based on the Air Quality Directive (2008/50/EC) and therefore they are always measured at the air quality stations. Even though there was more NO$_x$ (NO + NO$_2$)



and PM$_{2.5}$ data available, in this study we used only NO$_x$ and PM$_{2.5}$ data that was measured during the same period as BC was measured in order to make the comparison and trend analysis systematic.

The PM$_{2.5}$ concentration was measured with various different instruments, which are listed for each station in Table S1. The
instruments were based on four different methods: 1) attenuation of β-radiation (Thermo model FH 62 I-R); 2) tapered element oscillating microbalance (TEOM; Thermo different models); 3) optical detection (Grimm 180); and 4) collecting the particles in a cascade impactor and manually weighting the collected particles. The instruments, which use the methods 1-3, measure the concentration continuously. At RB2, where the PM$_{2.5}$ concentration was measured by collecting the particles in a cascade impactor and weighting the collected particles about three times a week so the time interval of these measurements varied from
two to three days. To compare the PM$_{2.5}$ measurements to BC concentration at RB2, the PM$_{2.5}$ concentration was interpolated to match BC timestamps.

NO$_x$ mass concentration [NO$_x$] was derived from the measurements of NO and NO$_2$ so that

$$[NO_x] = 1.533 \cdot [NO] + [NO_2]. \tag{1}$$

The mass concentration of NO and NO$_2$ were measured by instruments, which are based on the chemiluminescence method. The instruments used at each station are listed in Table S1.

## 2.4 Data processing

The data quality were assured by the data producer and invalid data were omitted from further analysis. At TR1, all the PM$_{2.5}$
data between 8 – 9 a.m. and 9 - 10 p.m. had to be omitted due to technical issues caused by the disturbance of the air conditioning. The concentrations were converted to ambient outdoor temperature and to normal atmospheric pressure (1013.25 hPa). In this study, we used one-hour averages for all the variables (only exception is the PM$_{2.5}$ data at RB2). The hourly mean values were calculated if the hour had at least 75 % of valid data. The hour of day always refers to the local time and the time stamp of the measurements are reported in the middle of the averaging period.

### 2.4.1 Trend analysis

We used seasonal Mann-Kendall test and Sen's slope estimator (Gilbert, 1987) in determining the statistical significances and the slopes of the long-term trends. Mann-Kendall test and Sen's slope estimator are non-parametric statistical methods, which allow missing data points. The method determines the trends for each season (here we used months) separately and tests if the
trends for different seasons are homogeneous. We used monthly median values in the trend analysis and we required at least 14 days of valid data for each month, otherwise the month was not taken into account in the trend analysis. Similar analysis



has been used in various trend studies (e.g., Collaud Coen et al., 2007; Collaud Coen et al., 2013; Li et al., 2014; Zhao et al., 2017).

## 3 Results and discussion

### 3.1 Spatial variation

The statistics of eBC, $PM_{2.5}$, and $NO_x$ concentrations from each site are presented in Figure 2. The figure includes all the data and the statistics were determined by using the 1 h mean values. Figure 2 shows that the arithmetic mean values differ from the median values, which means that the data of the air pollutants is not normally distributed at any station and that the data is skewed to the right (i.e., small concentrations occur more often and therefore the median is smaller than the mean).

As expected, the highest mean eBC concentrations were observed at the TR sites, where the mean eBC concentration varied from 0.77 to 2.08 µg m$^{-3}$ measured at the TR6 and TR3, respectively. At the DH sites, the mean eBC concentration varied from 0.64 to 0.80 µg m$^{-3}$, which were rather similar mean values as observed at the TR1, TR5, and TR6 (0.84, 0.83, and 0.77 µg m$^{-3}$, respectively). At the UB sites, the mean eBC concentrations were around 0.52 µg m$^{-3}$, which were clearly lower than at the TR and DH sites. The lowest mean eBC concentrations (0.28 µg m$^{-3}$) were observed, as expected, at the RB sites that had no local BC sources in vicinity.

Previous studies have shown that in addition to the traffic count, the BC concentration near traffic lines depends on various factors: the distance to the traffic lines (Enroth et al., 2016; Massoli et al., 2012; Zhu et al., 2002); the speed limit (Lefebvre et al., 2011); and the fraction of heavy-duty vehicles (Clougherty et al., 2013; Weichenthal et al., 2014). The surrounding buildings, vegetation and the wind conditions affect the dilution and therefore the BC concentrations as well (Abhijith et al., 2017; Brantley et al., 2014; Pirjola et al., 2012). Also, close by intersections may affect the BC concentrations if they induce traffic build-ups; BC emissions from vehicles that accelerate are higher than the emissions from a steadily moving vehicle (Imhof et al., 2005).

The abovementioned factors are probably the reason for the differences between the different TR stations and explain why the eBC concentration at TR3 was notably higher than at the other TR sites. TR3 was located in a street canyon right next to a very busy traffic line, which also has a rather high fraction of heavy-duty vehicles. According to Table 2, the traffic count on the closest street next to TR3 was around 44 400 vehicles per weekday and the fraction of heavy-duty vehicles was 14 %. The traffic count and fraction of heavy-duty vehicles is higher than for example on the street next to TR1 or at the street next to TR2. The area around TR3 consists also of a few busy intersections and the station is located in a street canyon, which probably increased the eBC concentrations even further.



The effects of total traffic count and traffic count of heavy-duty vehicles on the eBC concentration were studied in more detail in the supplementary material (Fig. S9), where the annual mean eBC concentrations were compared against the estimated weekday traffic counts of all vehicles and of heavy-duty vehicles in the nearest street or road. The annual means of eBC

concentration correlated better with the number of heavy-duty vehicles passing the station in a day ($R = 0.82$) than with the total traffic count during the day ($R = 0.71$). In general, the number of heavy-duty vehicles passing the TR1, TR5, and TR6 per day was low compared to TR2, TR3, and TR4, and this was also seen in the mean concentration of eBC. The effect of heavy-duty vehicles was expected, since the BC emissions from heavy-duty vehicles are higher compared to light-duty vehicles (Imhof et al., 2005).

It must be noted, that the distance from the stations to the nearest street varies, which affects the measured eBC concentration (Massoli et al., 2012; Zhu et al., 2002). A study by Enroth et al. (2016) estimated that eBC concentration decreases to a half at 33 m distance from the road. TR2 and TR3, where we observed rather high concentrations, were located right next to the street (in a 0.5 m distance), whereas the other stations had a longer distance to the nearest street (3 – 20 m). Previous studies have

also shown that tall trees in street canyons may deteriorate the air quality by preventing dispersion (Abhijith et al., 2017) and this may also have affected the higher measured concentrations at TR2 and TR3, since in Mäkelänkatu street there are two lines of trees framing the tram lines in middle the street (see Fig. S2b) and in Mannerheimintie street there are trees planted between the traffic lines and pavements (see Fig. S2c).

Higher concentrations of eBC than observed at this study have been reported at other urban areas in Europe. For example, Becerril-Valle et al. (2017) reported mean eBC values of 3.7 µg m$^{-3}$ at a traffic site and 2.33 µg m$^{-3}$ at an urban background site measured in Madrid in 2015. Singh et al. (2018) reported on average eBC concentrations of 1.83 and 1.34 µg m$^{-3}$ measured at several urban center and urban background sites in the United Kingdom during 2009–2011. Krecl et al. (2017) observed mean eBC concentration of 2.1 µg m$^{-3}$ at a street canyon site in Stockholm during weekdays in spring 2013. The BC

concentrations reported by other studies at different environments are higher than the eBC concentrations measured at corresponding environments around the HMA. Generally, the air quality in the HMA is good due to its coastal location, which enhances the dilution, and relatively small size, since the area is not as densely populated than other European capitals.

In addition to eBC, we also studied the variation of NO$_x$ and PM$_{2.5}$, which were measured at all of the stations. The spatial

variation of NO$_x$ was partly similar to that of eBC; the highest concentrations were measured at the TR sites and the lowest at the RB sites. At the TR sites, the mean concentrations varied between 44 – 147 µg m$^{-3}$ (lowest concentration at TR5 and highest at TR3) and the variation between the stations was rather similar to the variation of eBC. Like BC, NO$_x$ is highly dependent on the traffic related parameters such as the traffic count, the fraction of heavy traffic, the speed limit etc., which explain the similar variation observed. For the RB sites, however, the mean NO$_x$ (2 µg m$^{-3}$) was relatively low compared to





eBC at RB sites. Another difference to eBC was that, the $NO_x$ concentration at the DH sites was relatively lower, which was expected, since the $NO_x$ emissions from residential wood combustion are low. The correlation between eBC and $NO_x$ concentrations at each station are presented in Fig. S7a. As the likeness in the spatial variability already suggested, the eBC and $NO_x$ had rather similar sources and therefore they were expected to correlate. The correlation coefficient ($R$) between these

variables was the highest at the TR stations ($0.80 \leq R \leq 0.90$) and lower at the DH stations ($0.63 \leq R \leq 0.73$). At the background sites, the correlation coefficient varied more ($0.55 \leq R \leq 0.83$).

For the $PM_{2.5}$ there were not as clear patterns between different station categories as there were for eBC and $NO_x$. At the traffic sites the mean $PM_{2.5}$ concentration varied from 5.6 to 11.3 $\mu gm^{-3}$ (at TR6 and TR3, respectively) and at the detached housing

sites the variation was rather similar: from 5.6 to 11.3 $\mu gm^{-3}$ (at DH4 and DH2, respectively). For $PM_{2.5}$, the mean concentration at the background sites were not as clearly lower as it was for eBC and $NO_x$. These results show that $PM_{2.5}$ did not depend on local primary pollution sources as much as eBC or $NO_x$ did. $PM_{2.5}$ includes all different kind of aerosol particles, especially secondary aerosol, which may be anthropogenic or biogenic origin. In this size range, non-anthropogenic particles (e.g., secondary particles of biogenic origin; Dal Maso et al., 2005) are also contributing. The differences in the sources of eBC and

$PM_{2.5}$ concentrations were also seen in the correlation between these two variables; the $R$ between these two variables at different stations were notably lower ($0.36 \leq R \leq 0.67$) than the $R$ between eBC and $NO_x$ concentrations (Fig. S7b).

The fraction of eBC in the $PM_{2.5}$ is shown in Fig. 3 and it represent how big fraction of $PM_{2.5}$ consists of eBC. eBC was measured mostly in $PM_1$, since most of the BC particles are smaller than 1 $\mu$m in diameter (e.g., Enroth et al., 2016). Higher

eBC/$PM_{2.5}$ ratio indicates that there is a larger fraction of PM related to combustion sources. The highest median eBC/$PM_{2.5}$ ratio were observed at the TR sites where the ratio varied from 10 to 15 %. The second highest median ratios were observed at the DH sites and at UB1 where the ratios varied from 5 to 9 %. At the RB sites, the median fractions were the smallest: less than 5 %.

The results of the spatial variation show that eBC concentration and eBC/$PM_{2.5}$ ratio depend greatly on the distance to the pollution sources, which are, in this case, traffic and wood burning. The $NO_x$ is very dependent on the distance to the traffic sources only, since $NO_x$ is not significantly affected by residential wood burning. Since the $PM_{2.5}$ has many sources and generally high background levels, it is the least dependent component on the contribution of the local sources.

It must be noted, that the Figure 2 and 3 include all the data that was collected from 2009 to 2018 and that the sizes of the data sets for each station differ. Therefore, the year-to-year variation caused by the meteorological conditions and changes in the emission rates might have affected these results as some sites contained only one year of data (all DH sites and TR4).





### 3.3 Temporal variation

### 3.3.1 Long-term trends

A quick look in Table 1 already showed that the annual eBC mean values had seemingly decreased. To see if the decreasing eBC trend had a statistical significance, we applied the seasonal Kendal test (see Sect. 2.4.1) to the data sets that were at least

four-year-long (TR1, TR2, RB2, and UB1). Even though there was four-years of measurements at the UB2, it was omitted from this analysis, since the data availability at UB2 in 2016 – 2017 was not good enough. The seasonal Kendal test was applied to monthly medians values, which are presented in Fig. 4 for the eBC concentration at TR1, TR2, UB1, and RB2. We could not apply the trend analysis to any of the DH stations, since none of the DH stations had more than one year of eBC measurements.

A statistically significant (p-value < 0.05) decreasing trend was observed for all of the stations, which had at least four years of data (TR1, TR2, UB1, and RB2) as shown in Fig. 4 and in Table 3. The smallest absolute decrease was observed at the background stations UB1 and RB2, where the slopes of the trends were -0.02 and -0.01 $\mu$g m$^{-3}$ yr$^{-1}$, respectively. At TR1 the concentration decreased more rapidly by -0.04 $\mu$g m$^{-3}$ yr$^{-1}$, and at TR2 of the decrease was even greater: -0.09 $\mu$g m$^{-3}$ yr$^{-1}$. In

addition to the absolute trend, we also determined the relative trends by dividing the absolute slope of the trend by the overall median concentration. At TR1, UB1, and RB2 the relative trends were rather similar: -6.4, -5.9, and -7.8 % yr$^{-1}$, respectively. At TR2, the decrease was relatively faster: -10.4 % yr$^{-1}$.

To see how the decrease in eBC concentration compares to the trends of other air pollutants, we conducted the trend analysis

also for the PM$_{2.5}$ and NO$_x$ data. The resulted trends are also presented in Table 3. The only parameter for which we did not observe a statistically significant decreasing trend was PM$_{2.5}$ at TR2 (p-value = 0.05). The relative trends varied from station to station, but a common trait was that the concentrations of eBC and NO$_x$ decreased relatively faster than the concentration of PM$_{2.5}$. The trends of NO$_x$ concentration varied from -19.7 % yr$^{-1}$ (TR3) to -4.0 % yr$^{-1}$ (RB2) and the trends of PM$_{2.5}$ concentration varied from -3.9 % yr$^{-1}$ (UB1) to -2.7 % yr$^{-1}$ (RB2). Since there was a notable decrease in the eBC concentration

at TR3 between the years 2010 and 2015 (Table 1), the trend at TR3 was also studied, which is presented in Sect. S5. The analysis showed that the concentrations of eBC, NO$_x$, and PM$_{2.5}$ decreased about -12.2, -8.2, and -5.6 % yr$^{-1}$, respectively. So also at the TR3, the PM$_{2.5}$ concentration decreased relatively the slowest.

The concentrations of eBC and NO$_x$ are more sensitive to the changes in the traffic, since they are more dependent on the local

sources than PM$_{2.5}$ as discussed in the Sect. 3.2. Since PM$_{2.5}$ is not limited only to changes in primary traffic related emissions, it explains why the slope of the trend for PM$_{2.5}$ was relatively smaller than that of eBC and NO$_x$. In other words, since the pollutant emissions from traffic sources have generally decreased, it clearly affects the trends of eBC and NO$_x$, which are originating from local traffic sources, but to a lesser extent the trend of PM$_{2.5}$.





For eBC and $NO_x$ it is difficult to say which one of the pollutants decreased at faster rate. Their relative decreases were rather similar at TR1 and at UB1. At RB2, the decrease in eBC concentration was a more notable if compared to the decrease in $NO_x$ concentration. However, at RB2, there were no local sources so the situation is probably rather complicated and depends on

the atmospheric chemistry and aging of the pollutants. At TR2, there was a large difference and $NO_x$ seemed to decrease at double rate compared to eBC, which could be caused, for example, by the fast renewal of the city bus fleet. According to the Helsinki Regional Transport Authority (HSL) in 2015 17 % of the HSL buses were Euro VI or Euro VI energy efficient and in 2018 the fraction had increased to 48 %. A study by Järvinen et al. (2019) showed that moving to Euro VI buses from enhanced environmentally friendly vehicles (EEV) efficiently decreases the $NO_x$ emissions. However, it must be noted that

here the short time series cause uncertainty to the trends and for example, the year-to-year variability caused by the meteorological conditions could cause apparent decrease in pollutant concentrations.

The trends at TR1 and UB1 were investigated in more detail, since these stations had the longest time series and they were located closer to local sources in the HMA. To see if the traffic related emissions affected the trends, the trend analysis was

conducted separately for each hour so that the monthly median was determined for each hour of the day. Only the data from weekdays was included in the analysis. This trend analysis revealed that there were clear decreases in the eBC and $NO_x$ concentrations around the morning rush hour as shown in Fig. 5. In principle, this indicates that the primary eBC and $NO_x$ emissions from traffic sources have decreased most prominently, i.e., changes in traffic regulations and technological advancements have decreased eBC and $NO_x$ emissions. Oppositely, the times in $PM_{2.5}$ peaks did not correspond directly to

traffic hours at either of the stations.

According to the report about traffic in Helsinki, the traffic volume in Helsinki increased a bit, by 0.4 % $yr^{-1}$ during the period 2006–2016 (Helsinki, 2017; in Finnish only). However, in general, the number of vehicles entering and exiting the city center (including TR1) decreased by 1.5 % $yr^{-1}$ and the number of vehicles entering and exiting the central urban area (including

TR2, TR3, and UB1) decreased by 0.8 % $yr^{-1}$ during the same period, which obviously decreases the emissions from traffic. The decreases in traffic rates were especially observed in the North-West part of the central urban area (e.g., Fig. S10c), which probably affected the trends observed at TR3. However, at Mäkelänkatu, which is the main street next to TR2, there was no clear decrease in the traffic rate (Fig. S10f). The renewal of bus and vehicle fleet as well as cleaner renewable fuels have been shown to decrease both BC and $NO_x$ emissions (Järvinen et al., 2019; Pirjola et al., 2019; Timonen et al., 2017) and this would

be expected to be seen in the ambient eBC concentrations in traffic environment.

Since the traffic rates did not decrease at all of the stations, it can be concluded that the decreasing trends were due to the emission reduction of traffic, taken place mostly due to the efficient regulation of diesel and gasoline car exhaust particle number. This regulation has enforced the use of diesel particulate filters (DPFs) in new diesel passenger cars and heavy-duty





diesel vehicles, reducing their BC and PM emissions even more than 90 % up to 99 % if compared to the diesel vehicles without DPF (Bergmann et al., 2009; Preble et al., 2015). However, also the increased use of biofuels and gas as vehicle fuels, increased share of electric vehicles as well as improvements of fuel economy in internal combustion engines affect the observed trends. E.g., the increase of fuel injection pressure in diesel combustion can improve the fuel economy of engines and

simultaneously decrease the BC emissions of engines (Lähde et al., 2011).

Decreases in the eBC concentration (or absorption coefficient) have also been observed in the Finnish arctic (Dutkiewicz et al., 2014; Lihavainen et al., 2015). In general declining trends in atmospheric aerosol particle number concentration and particulate material has been observed in various different types of environments in Europe (Asmi et al., 2013). Similar results

for the trend of eBC concentration have been reported at several cities and countries in Europe. In Stockholm, Sweden, Krecl et al. (2017), reported about 60 % reduction in eBC concentration in a busy street canyon between the years 2006 and 2013 (i.e., 7.5 % $yr^{-1}$). Singh et al. (2018) observed a statistically significant decreasing trend in eBC concentration at five measurements stations, which operated during 2009–2016 and were located in different types of environments in the United Kingdom. The trends varied from -0.09 $\mu g\ m^{-3}\ yr^{-1}$ (-4.7 % $yr^{-1}$) at an urban background site to -0.80 $\mu g\ m^{-3}\ yr^{-1}$ (-8.0 % $yr^{-1}$)

at a curbside station. A trend study based only London reported on average -0.59 $\mu g\ m^{-3}\ yr^{-1}$ (-11 % $yr^{-1}$) decrease in eBC concentration at three roadside sites for the period 2010–2014 (Font and Fuller, 2016). Kutzner et al. (2018) observed statistically significant decreasing BC (both eBC and elemental carbon) trends for the period of 2005–2014 at several sites in Germany. The trends at traffic sites varied from -0.31 to -0.15 $\mu g\ m^{-3}\ yr^{-1}$ and the trends at urban background sites varied from -0.03 to -0.02 $\mu g\ m^{-3}\ yr^{-1}$.

These studies reported higher absolute trends compared to the absolute trends observed at our study, which is probably due to higher eBC concentrations at these sites. However, the relative trends were rather similar. Other studies also observed steeper absolute trends at urban sites compared to background sites. In addition to eBC, Krecl et al. (2017) studied also the trend of $NO_x$, and Font and Fuller (2016) studied the trend of $PM_{2.5}$. Contradictory to our study, Krecl et al. (2017) did not observe a

decreasing trend for $NO_x$. Font and Fuller (2016) reported decreasing trends for $PM_{2.5}$ concentration, which were relatively similar to the trends of eBC concentration.

The abovementioned studies, which were conducted in urban environments, also linked the decrease in BC concentration to traffic regulations. For example: Krecl et al. (2017) drew a connection between the decreasing eBC concentration and the

renewal of the vehicle fleet; Singh et al. (2018) and Kutzner et al. (2018) proposed that the reductions in eBC were linked to the local and national air quality policies; and Font and Fuller (2016) suspected that the eBC concentration decreased due to effective filters in diesel vehicles.



### 3.3.2 Seasonal, weekly and diurnal variation of BC

The diurnal variation of eBC was investigated separately for the cold and the warm seasons. According to Fig. S1, the coldest 5 months typically extended from November to March and the warmest 5 months from May to September. April and October were omitted from this analysis as transition months. The seasonal dependencies for each station separately are presented in
Fig. S12.

The seasonal and diurnal variations of eBC were rather similar between the stations that belong in the same category (Fig. S13). Instead of studying the variation at each station separately, we determined a mean diurnal variation for different station categories to study the variation more generally. The figures were plotted by calculating the mean concentration each hour of
each day of the week for the cold and the warm seasons separately. All the available data were taken into account when the diurnal variation from different stations were averaged together.

The mean seasonal, weekly and diurnal variation of eBC for different station categories are presented in Fig. 6. At the TR1, TR2, TR3, and TR4 (i.e., TR1-4) the seasonal and diurnal variation was different from TR5 and TR6 (i.e., TR5-6) and therefore
the mean diurnal variations for these sets of stations were plotted separately in Figs. 6a and 6b. Fig. 6c presents the mean diurnal variation averaged over all the DH sites. Since UB1 had a notably longer time series compared to UB2, these two time series were not combined and only the diurnal variation from UB1 is used in Fig. 6d. The diurnal variation at RB1 was also presented without combining it with RB2 data in Fig. 6e.

Fig. 6 shows two common traits observed at all of the TR, DH, and UB sites during both seasons: 1) eBC concentration peak appeared each weekday morning around 8 a.m. because of the morning rush hour; and 2) the lowest eBC concentrations were measured each day during the night around 3 a.m. when there were not much anthropogenic activities. In addition to these common trends, each station category had their own traits in the seasonal, weekly, and diurnal variation. We also did a similar variation analysis for the eBC/PM$_{2.5}$ ratio presented in Fig. S14. In general the eBC/PM$_{2.5}$ ratio seems to follow the variation
of eBC (i.e., the eBC varies relatively more than PM$_{2.5}$).

At the TR1-4, the seasonal variation of eBC was not strong and in Fig. 6a the lines for the cold and the warm season follow each other. The lack of seasonal variation in traffic environment was also observed previously in the HMA (Helin et al., 2018; Teinilä et al., 2019) and elsewhere (Kutzner et al., 2018; Reche et al., 2011). During weekdays, the morning concentration
peak of eBC occurred around 8 a.m. and the afternoon peak around 4 p.m. The variation of eBC concentration correlated with the diurnal variation of the traffic counts (see examples of traffic rates from Mannerheimintie and Mäkelänkatu in Fig. S10). eBC concentration was notably lower during weekend, when the traffic rates were also lower.





During the warm season at TR1-4 sites, the morning eBC concentration peak was notably higher than during the cold season. Since seasonal variation in the traffic rates were not expected, the observation is probably explained by the variation in WS. WS is one of the most important meteorological parameter that affects the air pollution concentrations so that the concentrations decrease with increasing WS due to more effective dilution (Järvi et al., 2008; Teinilä et al., 2019). A study by

Teinilä et al. (2019) showed that the diurnal variation of WS depended on the season: in summer, the WS had its minimum around 6 a.m. and it reached its maximum around 3 p.m. with the growing boundary layer; in winter the WS had no variation whatsoever. Similar diurnal variation in WS was seen in our data (Fig. S2). Since the WS in the morning is typically lower during the warm season than during the cold season, the differences in the WS and the dilution could explain why the eBC concentrations peak during the warm season mornings.

The seasonal variation of eBC was the most pronounced at the DH sites, where the lowest concentrations occurred in June and July and the highest concentrations in December and January (Fig. S11g–k), which is probably explained by the residential wood combustion during the cold season. At the DH sites, the highest concentrations were measured during the evening (Fig. 6c), when people started to warm up their houses and saunas after the workday. This was different to other station categories,

where the morning concentration peak was higher or similar compared to the afternoon peak. Domestic wood combustion also increased the concentrations during the weekend and unlike at the TR and UB sites, the eBC concentration at the DH sites were rather similar compared to the weekdays.

The effect of residential wood combustion on the diurnal variation was also observed at TR5-6 (Fig. 6b), which were closer to

the detached housing areas than the other TR sites (see Fig. 1). The temporal variation at TR5-6 was a mix between the variation observed at the other traffic sites (TR1-4) and at the DH sites. At TR5-6, the concentration of eBC was notably higher in the evenings during the cold season compared to the warm season, which is similar to the DH stations. Also, the maximum of the afternoon concentration peak occurred around the same time as at the DH sites. However, the effect of traffic is seen in the afternoon peaks, since the peaks grew more rapidly around the afternoon rush hour. The traffic also affected the morning

concentration peak, which is similar to the afternoon concentration peak, whereas at the DH sites, the morning peak was notably lower. At TR5-6, the eBC concentration during weekend was lower than in weekdays, which was observed at the other TR sites as well. However, the difference was not as pronounced, which is again due to the effect of wood combustion (e.g., increased eBC concentration during Saturday evening).

In the cold season, the diurnal variation of eBC at the UB1 was rather similar to the diurnal variation at the TR1-4, with the rush hour peaks occurring in the morning and afternoon (Fig. 6d). During the warm season, however, the afternoon rush hour peak was missing. Also in weekends, the concentrations during the warm season were considerably lower. Since the UB1 was not in vicinity to pollution sources, the effect of the meteorological parameters became more important. In the daytime during the warm season, the pollutants were diluted in the convective and more windy (Fig. S2) boundary layer more effectively,





whereas during the cold season the pollutants accumulate in the boundary layer, if it is shallow and does not grow during the day (Pohjola et al., 2004).

At the RB sites, there was a notable seasonal variation as the eBC concentrations are higher during the cold season (Fig. S13n-o), since regional and long-range transported wood combustion emissions also elevates eBC concentrations at background sites in winter (Luoma et al., 2019). The diurnal and weekly variation, however, were not as clear (e.g., Fig. 6e). Diurnal variation was mainly caused by the variation in the convective boundary layer height that caused mixing and dilution and not by local anthropogenic sources, which is expected for a regional background station.

It is important to note that the measured variation in the concentration of BC is affected by both: the BC emissions, and the atmospheric processes determining the dilution of the BC. For instance, the diurnal variation of the eBC concentration at a traffic site is affected by traffic rate at the nearby streets, roads and highways, but it is affected also by changes of local weather conditions possibly having diurnal variation. Such atmospheric parameters are e.g., ambient temperature, prevailing wind direction, wind speed, and height of the boundary layer. Furthermore, especially the ambient temperature can directly affect the emissions also; e.g., slightly higher wintertime morning concentrations in detached housing areas can be due to the effects of ambient temperature to the cold start emissions of passenger cars.

**Conclusions**

This study analyzed the time series of atmospheric eBC concentrations in traffic, detached house, urban background and regional background environments measured in years 2007 – 2018 in the Helsinki metropolitan area (HMA). The overall mean eBC concentration varied in the range of 0.77 – 2.08 µg m$^{-3}$ at TR stations, 0.64 – 0.80 µg m$^{-3}$ at DH stations, and 0.51 – 0.53 µg m$^{-3}$ at UB stations. At the RB sites the mean eBC concentrations were the lowest, about 0.28 µg m$^{-3}$.

The study highlights the importance of local sources in respect of urban air quality; it shows that the local traffic and local wood burning in residential areas forms the main sources of atmospheric eBC in urban area studied here. The influence of traffic was clearly seen at the traffic stations (TR) and the effect of domestic wood combustion at the detached housing sites (DH). At the TR sites, which located closer to residential areas (TR5-6), the diurnal variation showed signs that the air quality in the area was affected both by traffic and domestic wood burning emissions. The concentrations at the TR sites reached their peak in the weekday mornings and stayed elevated until the afternoon rush hour has passed. In the evening, after the workday, the eBC concentrations reached their maxima at the DH sites, where the residents started to warm up their houses with wood combustion. At the DH sites, the eBC concentrations did not decrease during the weekend like the concentration decreased at the TR sites.





The trend analysis conducted in this study showed very clearly decreasing trends for atmospheric BC. Decreasing eBC concentration has two positive effects: 1) improved air quality and 2) decreased warming effect on the climate by light absorbing aerosols. The absolute trends of eBC concentration were most notable at the traffic sites, where the trends varied from -0.04 to -0.09 µg m$^{-3}$ yr$^{-1}$. At an UB station and at an RB station, the absolute trends were -0.02 and -0.01 µg m$^{-3}$ yr$^{-1}$,

respectively. The relative trends of eBC concentration varied between -10.6 – -5.7 % yr$^{-1}$, which was rather similar to the relative trends of NO$_x$ concentration, which varied between -19.7 – -4.0 % yr$^{-1}$. However, the relative trends of PM$_{2.5}$ were did not decrease as rapidly as for eBC and NO$_x$ and the relative trends of PM$_{2.5}$ varied between -3.9 – -2.7 % yr$^{-1}$. For the eBC and NO$_x$ the most notable decrease was observed for the hours of morning rush hour, when traffic has the biggest effect on the air quality. The difference between the relative trends and the most notable decrease during the morning rush hour indicate that

especially the emissions from traffic have decreased.

This study suggest that the development in vehicle exhaust particle mitigation has been successful, at least from the viewpoint of BC and NO$_x$ emissions. Simultaneously, this study clearly shows the need for regulation and mitigation of emissions from residential wood combustion, which is, according to the emission inventories, actually the most significant BC source in Finland (Rautalahti and Kupiainen, 2016). The research material of this study did not allow the assessment of long-term trends

in DH areas but, in principle, any similar technology changes have not been observed in wood combustion than has been in vehicles.

*Acknowledgements* This work was supported by the Academy of Finland Centre of Excellence in Atmospheric Science (grant no. 307331) and NANOBIOMASS (207537), ACTRIS-Finland (329274), the Regional innovations and experimentations funds (project HAQT, AIKO014) and Business Finland (BC Footprint -project 528/31/2019, and MegaSense Smart City-

project 6884/31/2018). In addition, the work was financially supported by European Commission through ACTRIS2 (654109) and ACTRIS-IMP (871115) and through SMart URBan Solutions for air quality, disasters and city growth (689443), ERA-NET-Cofund and by University of Helsinki (ACTRIS-HY).

*Data availability* All the data presented in this study is open access. The air quality data collected by the HSY is available

from their website (https://www.hsy.fi/fi/asiantuntijalle/avoindata/Sivut/default.aspx, last access: 27 February 2020). The data collected from the SMEAR sites (UB2 and RB2) has been accessed by the Smart-SMEAR online tool (Junninen et al., 2009).

*Author contribution* Krista Luoma did the data analysis and wrote the manuscript. All the other authors helped with analysing the results and they also reviewed and commented the manuscript.



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

©c Author(s) 2020. CC BY 4.0 License.





Järvinen, A., Timonen, H., Karjalainen, P., Bloss, M., Simonen, P., Saarikoski, S., Kuuluvainen, H., Kalliokoski, J., Dal Maso, M., Niemi, J. V., and Rönkkö, T.: Particle emissions of Euro VI, EEV and retrofitted EEV city buses in real traffic, Environ. Pollut., 250, 708-716, 10.1016/j.envpol.2019.04.033, 2019.

Krecl, P., Johansson, C., Targino, A. C., Ström, J., and Burman, L.: Trends in black carbon and size-resolved particle number concentrations and vehicle emission factors under real-world conditions, Atmos. Environ., 165, 155-168, 10.1016/j.atmosenv.2017.06.036, 2017.

Krzyżanowski, M., Kuna-Dibbert, B., and Schneider, J.: Health effects of transport-related air pollution, WHO Regional Office Europe, 2005.

Kutzner, R. D., von Schneidemesser, E., Kuik, F., Quedenau, J., Weatherhead, E. C., and Schmale, J.: Long-term monitoring of black carbon across Germany, Atmos. Environ., 185, 41-52, 10.1016/j.atmosenv.2018.04.039, 2018.

Lefebvre, W., Fierens, F., Trimpeneers, E., Janssen, S., Van de Vel, K., Deutsch, F., Viaene, P., Vankerkom, J., Dumont, G., Vanpoucke, C., Mensink, C., Peelaerts, W., and Vliegen, J.: Modeling the effects of a speed limit reduction on traffic-related elemental carbon (EC) concentrations and population exposure to EC, Atmos. Environ., 45, 197-207, 10.1016/j.atmosenv.2010.09.026, 2011.

Lelieveld, J., Evans, J. S., Fnais, M., Giannadaki, D., and Pozzer, A.: The contribution of outdoor air pollution sources to premature mortality on a global scale, Nature, 525, 367, 10.1038/nature15371, 2015.

Li, J., Carlson, B. E., Dubovik, O., and Lacis, A. A.: Recent trends in aerosol optical properties derived from AERONET measurements, Atmos. Chem. Phys., 14, 12271-12289, 10.5194/acp-14-12271-2014, 2014.

Lihavainen, H., Hyvärinen, A., Asmi, E., Hatakka, J., and Viisanen, Y.: Long-term variability of aerosol optical properties in northern Finland, Boreal Environ. Res., 20, 526-541, 2015.

Luoma, K., Virkkula, A., Aalto, P., Petäjä, T., and Kulmala, M.: Over a 10-year record of aerosol optical properties at SMEAR II, Atmos.
Chem. Phys., 19, 11363-11382, 10.5194/acp-19-11363-2019, 2019.

Lähde, T., Rönkkö, T., Happonen, M., Söderström, C., Virtanen, A., Solla, A., Kytö, M., Rothe, D., and Keskinen, J.: Effect of fuel injection pressure on a heavy-duty diesel engine nonvolatile particle emission, Environ. Sci. Technol., 45, 2504-2509, 10.1021/es103431p, 2011.

Massoli, P., Fortner, E. C., Canagaratna, M. R., Williams, L. R., Zhang, Q., Sun, Y., Schwab, J. J., Trimborn, A., Onasch, T. B., Demerjian, K. L., Charles, E., Worsnop, D., and Jayne, J.: Pollution gradients and chemical characterization of particulate matter from vehicular traffic
near major roadways: Results from the 2009 Queens College Air Quality Study in NYC, Aerosol Sci. Technol., 46, 1201-1218, 10.1080/02786826.2012.701784, 2012.

Pakkanen, T. A., Kerminen, V.-M., Ojanen, C. H., Hillamo, R. E., Aarnio, P., and Koskentalo, T.: Atmospheric black carbon in Helsinki, Atmos. Environ., 34, 1497-1506, 10.1016/S1352-2310(99)00344-1, 2000.

Petzold, A. and Schönlinner, M.: Multi-angle absorption photometry—a new method for the measurement of aerosol light absorption and
atmospheric black carbon, J. Aerosol Sci, 35, 421-441, 10.1016/j.jaerosci.2003.09.005, 2004.

Petzold, A., Ogren, J. A., Fiebig, M., Laj, P., Li, S.-M., Baltensperger, U., Holzer-Popp, T., Kinne, S., Pappalardo, G., Sugimoto, N., Wehrli, C., Wiedensohler, A., and Zhang, X.-Y.: Recommendations for reporting "black carbon" measurements, Atmos. Chem. Phys., 13, 8365-8379, 10.5194/acp-13-8365-2013, 2013.

Pirjola, L., Lähde, T., Niemi, J., Kousa, A., Rönkkö, T., Karjalainen, P., Keskinen, J., Frey, A., and Hillamo, R.: Spatial and temporal
characterization of traffic emissions in urban microenvironments with a mobile laboratory, Atmos. Environ., 63, 156-167, 10.1016/j.atmosenv.2012.09.022, 2012.

Pirjola, L., Niemi, J. V., Saarikoski, S., Aurela, M., Enroth, J., Carbone, S., Saarnio, K., Kuuluvainen, H., Kousa, A., Rönkkö, T., and Hillamo, R.: Physical and chemical characterization of urban winter-time aerosols by mobile measurements in Helsinki, Finland, Atmos. Environ., 158, 60-75, 10.1016/j.atmosenv.2017.03.028, 2017.



Pirjola, L., Kuuluvainen, H., Timonen, H., Saarikoski, S., Teinilä, K., Salo, L., Datta, A., Simonen, P., Karjalainen, P., Kulmala, K., and Rönkkö, T.: Potential of renewable fuel to reduce diesel exhaust particle emissions, Appl. Energ., 254, 113636, 10.1016/j.apenergy.2019.113636, 2019.

Pohjola, M. A., Rantamäki, M., Kukkonen, J., Karppinen, A., and Berge, E.: Meteorological evaluation of a severe air pollution episode in Helsinki on 27-29 December 1995, Boreal Environ. Res., 9, 75-87, 2004.

Preble, C. V., Dallmann, T. R., Kreisberg, N. M., Hering, S. V., Harley, R. A., and Kirchstetter, T. W.: Effects of particle filters and selective catalytic reduction on heavy-duty diesel drayage truck emissions at the Port of Oakland, Environ. Sci. Technol., 49, 8864-8871, 10.1021/acs.est.5b01117, 2015.

Rautalahti, E. and Kupiainen, K.: Emissions of black carbon and methane in Finland, Ministry of the Environment, Helsinki, 2016.

Reche, C., Querol, X., Alastuey, A., Viana, M., Pey, J., Moreno, T., Rodríguez González, S., González Ramos, Y., Fernández-Camacho, R., de la Rosa, J., Dall'Osto, M., Prévôt, A. S. H., Hueglin, C., Harrison, R. M., and Quincey, P.: New considerations for PM, black carbon and particle number concentration for air quality monitoring across different European cities, Atmos. Chem. Phys., 11, 6207–6227, 10.5194/acp-11-6207-2011, 2011.

Rönkkö, T. and Timonen, H.: Overview of sources and characteristics of nanoparticles in urban traffic-influenced areas, J. Alzheimer's Dis., 72, 15-28, 10.3233/JAD-190170, 2019.

Savolahti, M., Karvosenoja, N., Tissari, J., Kupiainen, K., Sippula, O., and Jokiniemi, J.: Black carbon and fine particle emissions in Finnish residential wood combustion: Emission projections, reduction measures and the impact of combustion practices, Atmos. Environ., 140, 495-505, 10.1016/j.atmosenv.2016.06.023, 2016.

Singh, V., Ravindra, K., Sahu, L., and Sokhi, R.: Trends of atmospheric black carbon concentration over the United Kingdom, Atmos. Environ., 178, 148-157, 10.1016/j.atmosenv.2018.01.030, 2018.

Stocker, T. F., Qin, D., Plattner, G.-K., Tignor, M., Allen, S. K., Boschung, J., Nauels, A., Xia, Y., Bex, V., and Midgley, P. M.: Climate change 2013: The physical science basis, Cambridge University Press, 2013.

Teinilä, K., Aurela, M., Niemi, J. V., Kousa, A., Petäjä, T., Järvi, L., Hillamo, R., Kangas, L., Saarikoski, S., and Timonen, H.: Concentration variation of gaseous and particulate pollutants in the Helsinki city centre — observations from a two-year campaign from 2013–2015, Boreal Environ. Res., 24, 115-136, 2019.

Timonen, H., Karjalainen, P., Saukko, E., Saarikoski, S., Aakko-Saksa, P., Simonen, P., Murtonen, T., Dal Maso, M., Kuuluvainen, H., Bloss, M., Ahlberg, E., Svenningsson, B., and Pagels, J.: Influence of fuel ethanol content on primary emissions and secondary aerosol formation potential for a modern flex-fuel gasoline vehicle, Atmos. Chem. Phys., 17, 5311-5329, 10.5194/acp-17-5311-2017, 2017.

Timonen, H., Karjalainen, P., Aalto, P., Saarikoski, S., Mylläri, F., Karvosenoja, N., Jalava, P., Asmi, E., Aakko-Saksa, P. i., Saukkonen, N., Laine, T., Saarnio, K., Niemelä, N., Enroth, J., Väkevä, M., Oyola, P., Pagels, J., Ntziachristos, L., Cordero, R., Kuittinen, N., Niemi, J., and Rönkkö, T.: Adaptation of black carbon footprint concept would accelerate mitigation of global warming, Environ. Sci. Technol., 53 (21), 12153-12155, 10.1021/acs.est.9b05586, 2019.

Vallius, M. J., Ruuskanen, J., Mirme, A., and Pekkanen, J.: Concentrations and estimated soot content of PM1, PM2. 5, and PM10 in a subarctic urban atmosphere, Environ. Sci. Technol., 34, 1919-1925, 10.1021/es990603e, 2000.

Weichenthal, S., Farrell, W., Goldberg, M., Joseph, L., and Hatzopoulou, M.: Characterizing the impact of traffic and the built environment on near-road ultrafine particle and black carbon concentrations, Environ. Res., 132, 305-310, 10.1016/j.envres.2014.04.007, 2014.

Virkkula, A., Mäkelä, T., Hillamo, R., Yli-Tuomi, T., Hirsikko, A., Hämeri, K., and Koponen, I. K.: A simple procedure for correcting loading effects of Aethalometer data, J. Air Waste Ma., 57, 1214-1222, 10.3155/1047-3289.57.10.1214, 2007.



Zhao, B., Jiang, J. H., Gu, Y., Diner, D., Worden, J., Liou, K.-N., Su, H., Xing, J., Garay, M., and Huang, L.: Decadal-scale trends in regional aerosol particle properties and their linkage to emission changes, Environ. Res. Lett., 12, 054021, 10.1088/1748-9326/aa6cb2, 2017.

Zhu, Y., Hinds, W. C., Kim, S., Shen, S., and Sioutas, C.: Study of ultrafine particles near a major highway with heavy-duty diesel traffic, Atmos. Environ., 36, 4323-4335, 10.1016/S1352-2310(02)00354-0, 2002.





**Table 1: Annual mean values of eBC concentration for each station in units of µg m⁻³. Color-coding indicates the magnitude of the mean concentration; darker color refers to higher concentration. The annual means at UB2 in 2016 – 2017 are bracketed, since there was less than 50 % of valid data.**

| Station | 2007 | 2008 | 2009 | 2010 | 2011 | 2012 | 2013 | 2014 | 2015 | 2016 | 2017 | 2018 |
|---|---|---|---|---|---|---|---|---|---|---|---|---|
| TR1 | | | | | 1.27 | | 0.90 | 0.81 | 0.72 | 0.80 | 0.68 | 0.73 |
| TR2 | | | | | | | | | 1.35 | 1.24 | 1.08 | 0.99 |
| TR3 | | | | 2.64 | | | | | 1.55 | | | |
| TR4 | | | | | | 1.58 | | | | | | |
| TR5 | | | | | | | | 0.91 | | 0.76 | | 0.83 |
| TR6 | | | | | | | | | 0.88 | | 0.67 | |
| DH1 | | | 0.80 | | | | | | | | | |
| DH2 | | | | | | | | 0.80 | | | | |
| DH3 | | | | | | | | | | 0.65 | | |
| DH4 | | | | | | | | | | | 0.64 | |
| DH5 | | | | | | | | | | | | 0.74 |
| UB1 | | | | | | 0.68 | 0.59 | 0.53 | 0.52 | 0.50 | 0.42 | 0.49 |
| UB2 | | | | | | | | | 0.54 | (0.56) | (0.44) | 0.50 |
| RB1 | | | | | | | | | | 0.26 | | 0.29 |
| RB2 | 0.32 | 0.41 | 0.37 | 0.43 | 0.37 | | 0.21 | 0.25 | 0.21 | 0.19 | 0.16 | 0.19 |





**Table 2: The traffic counts at the nearest streets to the TR stations. The traffic counts and the fraction of the heavy-duty are from from the yearly Helsinki traffic reports. The traffic rates are given for working days only. The streets and roads mentioned here are marked in the Figs. S3.**

| Station | Street name | Traffic count (vehicles/weekday) | Heavy-duty (%) | Distance to street edge (m) | Reference year |
|---------|-------------|-----------------------------------|-----------------|------------------------------|-----------------|
| TR1 | Mannerheimintie | 15 800 | 11 | 3 | 2017 |
| | Kaivokatu | 20 100 | 8 | 40 | 2017 |
| TR2 | Mäkelänkatu | 28 100 | 11 | 0.5 | 2017 |
| TR3 | Mannerheimintie | 44 400 | 14 | 0.5 | 2010 |
| | Reijolankatu | 19 400 | - | 25 | 2010 |
| TR4 | Kehä I | 69 200 | 8 | 5 | 2012 |
| | Tattariharjuntie | 13 700 | 13 | 120 | 2012 |
| TR5 | Tikkurilantie | 9 500 | - | 7 | 2016 |
| TR6 | Turuntie | 29 300 | 4 | 20 | 2017 |
| | Lintuvaarantie | 15 400 | 5 | 30 | 2017 |
| | Kehä I | 68 900 | 4 | 250 | 2017 |





**Table 3: Results of the trend analysis of the eBC, PM$_{2.5}$, and NO$_x$ concentrations. The values without brackets are the absolute trends in units of µg yr$^{-1}$, the values in the square brackets are the 5$^{th}$ and 95$^{th}$ uncertainty limits of the trend in units of µg yr$^{-1}$, and the bracketed values are the relative trends in units of % yr$^{-1}$. The values for PM$_{2.5}$ at TR2 are italicized, since they were statistically not significant (p-value = 0.05).**

| Station | eBC (µg yr$^{-1}$) | PM$_{2.5}$ (µg yr$^{-1}$) | NO$_x$ (µg yr$^{-1}$) | Measurement years |
|---|---|---|---|---|
| **TR1** | -0.04 [-0.06; -0.02] (-6.5 % yr$^{-1}$) | -0.24 [-0.38; -0.01] (-3.7 % yr$^{-1}$) | -3.11 [-4.18; -1.87] (-7.1 % yr$^{-1}$) | 2011, 2013 – 2018 |
| **TR2** | -0.09 [-0.11; -0.05] (-10.6 % yr$^{-1}$) | *-0.46* *[-0.71; 0.02]* *(-7.1 % yr$^{-1}$)* | -11.00 [-12.47; -8.70] (-19.7 % yr$^{-1}$) | 2015 – 2018 |
| **UB1** | -0.02 [-0.03; -0.01] (-5.7 % yr$^{-1}$) | -0.20 [-0.34; -0.05] (-3.9 % yr$^{-1}$) | -0.80 [-1.08; -0.50] (-5.0 % yr$^{-1}$) | 2012 – 2018 |
| **RB2** | -0.01 [-0.02; -0.01] (-7.6 % yr$^{-1}$) | -0.10 [-0.20; -0.03] (-2.7 % yr$^{-1}$) | -0.05 [-0.08; -0.03] (-4.0 % yr$^{-1}$) | 2006 – 2018 |





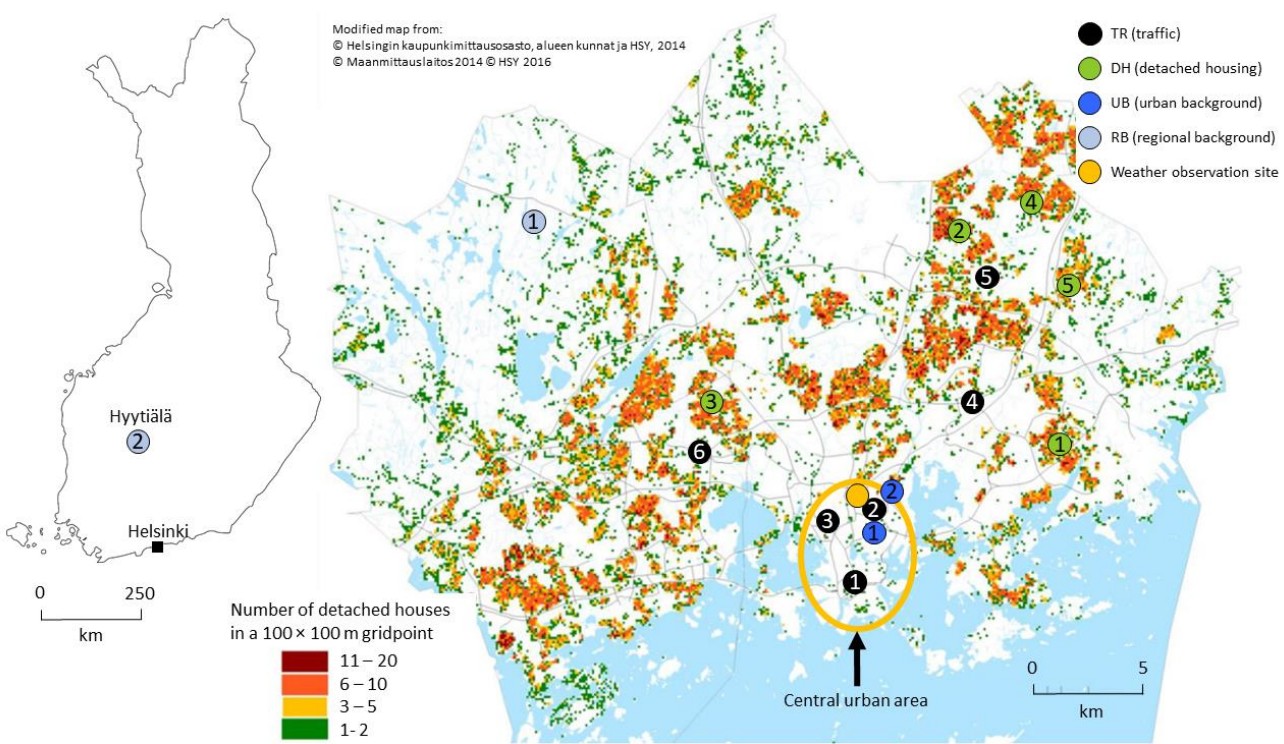

Figure 1: The locations of the stations and the density of the detached houses. Differently colored markers indicate different station categories. The central urban area is marked to the map with an orange circle.



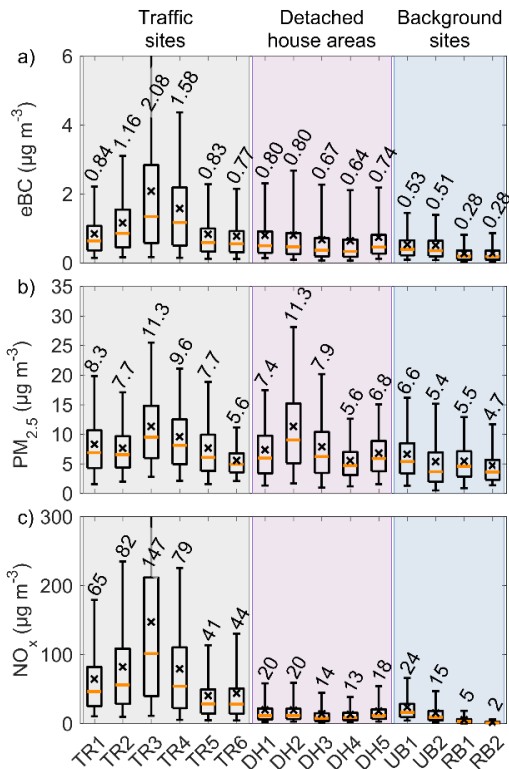

**Figure 2: The statistics of the a) eBC, b) PM$_{2.5}$, and c) NO$_x$ concentrations at each station. The boxplots are presented for 1 h mean values. The orange line in the middle of each box represents the median, the edges of the boxes represent the 25$^{th}$ and 75$^{th}$ percentiles, and the whiskers represent the 5$^{th}$ and 95$^{th}$ percentiles. The black cross is the arithmetic mean, and its numerical value is reported above or below each box. The background color represents the station type: gray for traffic sites, purple for the detached housing sites, and blue for the background sites. The 75$^{th}$ percentiles of eBC and NO$_x$ concentration at TR3, which are not visible at the figure, were 6.7 and 440 µg m$^{-3}$, respectively.**

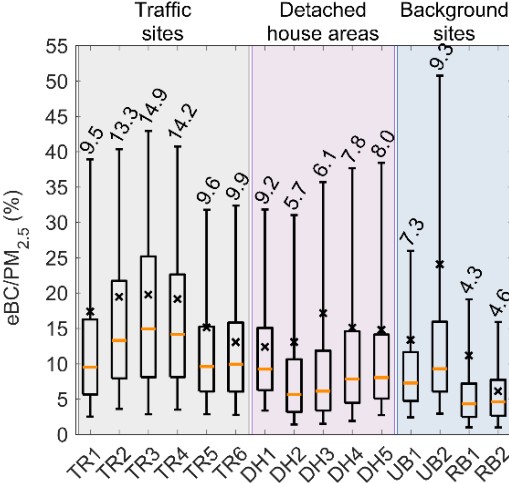

**Figure 3: The statistics of eBC/PM2.5 fraction at each station. The explanation for the markers is the same as in Fig. 2, except here the values reported above each box are the median values.**





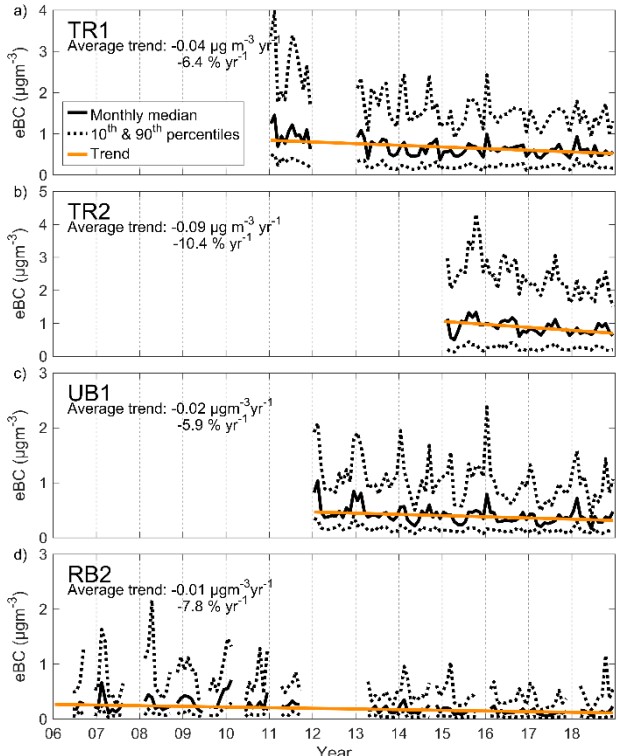

**Figure 4: Time series and the trends of eBC concentration at TR1, TR2, UB1, and RB2. The solid black line represents the monthly medians, the dashed lines represent the 10th and 90th monthly percentiles, and the orange line is the fitted long-term trend.**

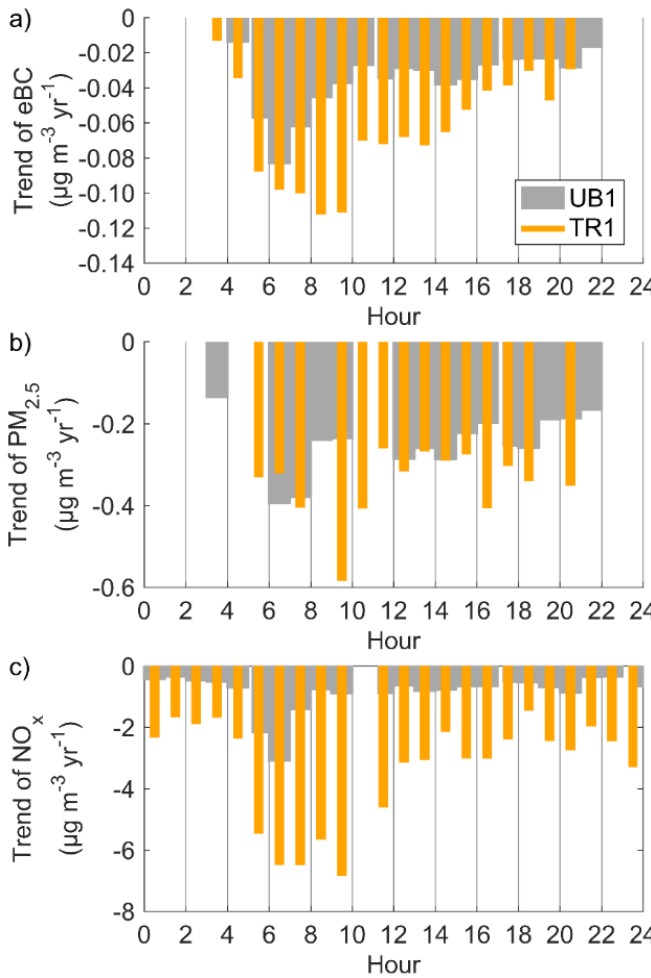

**Figure 5: Trends for the hourly data. Here, only data from weekdays were used. None of the pollutants had a positive trend for any time of the day.**

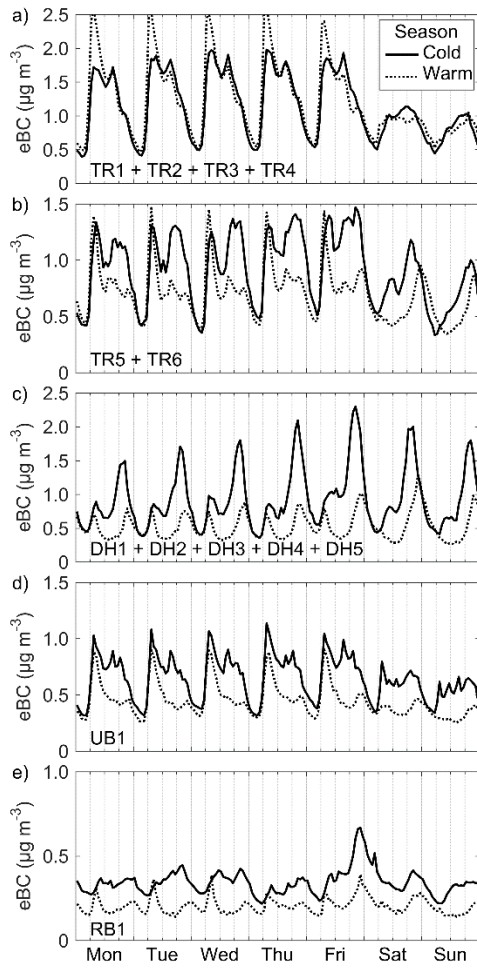

**Figure 6: Diurnal variation of eBC for different station categories: a) traffic sites that are not influenced by wood burning (TR1 – TR4), b) traffic sites that are influenced by wood burning (TR5 – TR6), c) detached housing sites (DH1 – DH5), d) urban background (UB1), and e) regional background (RB1). The diurnal variation is determined separately for the cold (from November to March) and warm (from May to September) periods.**