# Peer review of "Spatiotemporal variation and trends of equivalent black carbon in the Helsinki metropolitan area in Finland"

_Atmospheric Chemistry and Physics, 2020_

## Referee Comment (RC1) · Anonymous Referee #2 · 20 May 2020

The overall quality of the preprint is excellent; any comments regarding the scientific aspects are minor.

Equivalent BC: Is there a different conversion for optical to mass concentration of BC from vehicle emissions vs. wood combustion?

Conclusions: Can you make any projections for the future of vehicle related BC emissions considering future changes in regulation, changes in traffic patterns, and changes in vehicle fleet type, either through better combustion, emission control or changing energy sources, e.g. electric?

Typographical and Formatting:

[Figure]

The abstract could be shortened and streamlined, see supplemental pdf file suggestions.

In presenting ranges throughout the MS, e.g. page 13 line 18, use that format consistently to avoid confusion between minus sign and dash, "The trends at traffic sites varied from -0.31 to -0.15 $\mu$g m-3 yr-1 and ... "

Page 13, line15 at a curbside station. A trend study based only in London

Page 17 line 2 concentration has two positive effects: 1) improved air quality and 2) decreased warming effect on the global climate by light absorbing

Page 17 Line 6 relative trends of NOx concentration, which varied between -19.7 – -4.0 % yr-1. However, the relative trends of PM2.5 were did not decrease as rapidly as for eBC and NOx and the relative trends of PM2.5 varied between -3.9 – -2.7 % yr-1.

Fig. 5 Annual trends for the hourly data ...

Fig. S6 Add R value of linear fit. Add confidence limits, of slope, ie., is it significantly different than 1.0? The "trouble" with having so many data points is that one can see only the outline and not the distribution in the densest regions. To help the visual effect, plot the points in a lighter grey with the fits and limits in more intense, overlying colors.

Fig. S7 Add R value and statistics.

Fig. S9 Notably R value seem to be much less than for S6 and S7 by my ocular analysis.

Please also note the supplement to this comment:
https://www.atmos-chem-phys-discuss.net/acp-2020-201/acp-2020-201-RC1-supplement.pdf

**Supplement:**

Abstract streamlining

Abstract: In this study, we present results of 12 years of black carbon (BC) measurements at 14  measurement sites around the Helsinki metropolitan area (HMA) and at one background site outside the HMA. The main local sources of BC in the HMA are traffic and residential wood combustion in fireplaces and sauna stoves. All the BC measurements were conducted optically and therefore we refer to the measured BC as equivalent BC (eBC). Measurement stations were located  indifferent  environments that represented traffic environments , detached housing areas , urban background , and regional background sites . The measurements of eBC were conducted during 2007 through 2018; however, the times  and the length of the  series varied by site . The largest annual mean eBC concentrations were measured at the traffic sites (0.67 – 2.64 $\mu$ g m-3) and the lowest at the regional background sites (0.16 – 0.29 $\mu$ g m-3). The annual mean eBC concentrations at the  housing sites varied from 0.64 – 0.80 $\mu$ g m-3 and the annual mean eBC concentrations at the urban background sites varied from 0.42 – 0.68 $\mu$ g m-3. The clearest seasonal variation was observed at the  housing sites, where  residential wood combustion increased the eBC concentrations during the cold season. Traffic rates and wood burning influenced  diurnal and weekly variations of eBC concentration in different types of environments. The dependency was not so clear for the other air pollutants, which were  NOx and mass concentration of particles smaller than 2.5 s. The dependenfour sites, which had at least four-year  time series available the eBC concentrations had statistically significant decreasing trends, which varied in the range of -10.4 to -5.9 % yr-1. Compared to  trends determined at the urban and regional background sites, the absolute trends decreased  fastest at  traffic sites  especially during  morning rush hour. Relative long-term trends of eBC and NOx were similar  and their concentrations decreased more rapidly than that of PM2.5. These results indicate that especially  emissions from traffic have decreased in the HMA during the last decade. This shows that air pollution control, new emission standards and a newer fleet of vehicles really have an effect on air quality.

---

## Referee Comment (RC2) · Anonymous Referee #3 · 6 Aug 2020

The article presents a long-term field study of BC, NOx and PM2.5 at different locations in Finland. Measurements at 4 locations allow for statistical evaluation of long-term trends, which show statistically significant reduction in BC and NOx. The study concludes that the new vehicle emission standards are responsible for the reduction of traffic emissions. The article is well written and can be published in ACP with minor revisions (see below).

1. As noted by the authors, the pollutant concentrations depend both on emission rates and atmospheric dilution. To support the author claims it is critical to quantitatively assess the long-term meteorological trends, especially for wind speed (during winter)

and nocturnal mixing height (during summer). The sensitivity of pollutant concentration to the meteorological parameters can be studied using a "BC versus wind speed plot" and "BC versus mixing height plot" (the same for NOx and PM2.5).

2. Authors claim that the detached housing areas are influenced by local wood-burning emissions, the assessment being supported by lower NOx/BC ratio at DH sites compared to TR sites. For the sites where Aethalometer was used (DH4 and DH5) the source contributions should be quantitatively assessed using wavelength dependence of the aerosol absorption (Sandradewi et al., 2008).

3. When comparing BC concentrations measured by MAAP at 637 nm and Aethalometer at 880 nm the wavelength dependence of the aerosol absorption should be taken into account (alternatively all measurements can be reported at the same wavelength). For the reader's convenience, please specify the measurement wavelength of the MAAP on Page 6 Line 12.

Reference

Sandradewi, J., Prévôt, A. S. H., Szidat, S., Perron, N., Alfarra, M. R., Lanz, V. A., Weingartner, E., and Baltensperger, U.: Using aerosol light absorption measurements for the quantitative determination of wood burning and traffic emission contributions to particulate matter, Environ. Sci. Technol., 42, 3316–3323, doi:10.1021/es702253m, 2008.
* * *

---

## Author Comment (AC1) · 5 Oct 2020

**Author's response to review**
Manuscript: acp-2020-201

Big thanks to the reviewers whose comments and suggestions improved the manuscript. We took the comments into account and did some changes in the manuscript. See below my point-by-point reply to the comments.

Best,

Krista Luoma

10   **Comments by the Referee #2:**

**Equivalent BC: Is there a different conversion for optical to mass concentration of BC from vehicle emissions vs. wood combustion?**

There was no different conversion factor (mass absorption or attenuation cross section) used at different sites. We added discussion about using the same constant MAC value at all the sites:

15   *As mentioned in the Sect. 2.2, we applied constant mass absorption cross section (MAC) values to convert the optically measured absorption data to eBC concentration. However, the MAC may vary depending on the chemical composition, shape and the mixing state of the PM. The MAC increases for aged BC particles, as the BC particles get coated with a scattering or slightly absorbing coating, which act as a lens increasing the absorption of the BC core (Lack and Cappa, 2010; Yuan et al., 2020). At TR sites, the freshly emitted BC particles from local traffic probably have no coating on the particles, but at the*

20   *remote sites, however, particles are carried over longer distances and the observed BC at these sites is more aged and likely more coated. Therefore, it is probable that the real MAC at the background sites was higher compared to TR sites. If the differences in the MAC values were taken into account, it could possibly increase the difference between the traffic and background sites. The source of the BC may also have an effect on the MAC, but for example Yuan et al. (2020) and Zotter et al. (2017) did not observe notable difference between the MAC for particles originating from traffic or wood combustion.  In*

25   *addition to spatial variation, the MAC can also vary temporally, which could affect the observed seasonal and diurnal variations and trends (presented in Sect. 3.3) as well. However, determining the variations of MAC would require extensive long-term measurements of chemical composition of the BC particles in different environments and therefore further analysis of the effect of MAC is omitted here.*

**Conclusions: Can you make any projections for the future of vehicle related BC emissions considering future changes in regulation, changes in traffic patterns, and changes in vehicle fleet type, either through better combustion, emission control orchanging energy sources, e.g. electric?**

A paragraph discussing the future changes in the air quality was added in the conclusions:

5 *This study suggests that the development in vehicle exhaust particle mitigation has been successful, at least from the viewpoint of BC and NOx emissions. With the current development, the pollution concentrations are expected to decrease in the next years as well. In general, the vehicle fleet is renewing, electric and hybrid cars are gaining popularity, and vehicles that run with biofuels or gas are becoming more common. The operator of public traffic in the HMA (HSL) aims to cut more than 90 % of their bus emissions (NOx, PM, CO2) by the year 2025 compared to the year 2010, which will improve the air quality*
10 *especially at the main roads in the HMA, where several bus lines operate.*

**The abstract could be shortened and streamlined, see supplemental pdf file suggestions.**

Thank you for the suggestions, the abstract was improved according to the supplement.

15 **In presenting ranges throughout the MS, e.g. page 13 line 18, use that format consistently to avoid confusion between minus sign and dash, "The trends at traffic sites varied from -0.31 to -0.15 µg m-3 yr-1 and ... "**

This issue was fixed and now the ranges of concentrations or trends are consisted (we used the "from … to …" format).

**Page 13, line 15 at a curbside station. A trend study based only in London**

20 This issue was fixed.

**Page 17 line 2 concentration has two positive effects: 1) improved air quality and 2) decreased warming effect on the global climate by light absorbing**

This issue was fixed.

25

**Page 17 Line 6 relative trends of NOx concentration, which varied between -19.7 – -4.0 % yr-1. However, the relative trends of PM2.5 were did not decrease as rapidly as for eBC and NOx and the relative trends of PM2.5 varied between -3.9 – -2.7 % yr-1.**

This issue was fixed.

30

**Fig. 5 Annual trends for the hourly data ...**

This issue was fixed.

**Fig. S6 Add R value of linear fit. Add confidence limits, of slope, ie., is it significantly different than 1.0? The "trouble" with having so many data points is that one can see only the outline and not the distribution in the densest regions. To help the visual effect, plot the points in a lighter grey with the fits and limits in more intense, overlying colors.**

This figure was changed so that the surface of the figure shows the number of data points in each grid point. The $R^2$ of the fit is presented in the figure and the standard errors of the fit are given in the caption.

**Fig. S7 Add R value and statistics.**

The $R$ values are now presented in the caption.

**Fig. S9 Notably R value seem to be much less than for S6 and S7 by my ocular analysis.**

I rechecked the calculations of the correlation parameter R, but the values were still the same.

**Comments by the referee #3:**

**The article presents a long-term field study of BC, NOx and PM2.5 at different locations in Finland. Measurements at 4 locations allow for statistical evaluation of long-term trends, which show statistically significant reduction in BC and NOx. The study concludes that the new vehicle emission standards are responsible for the reduction of traffic emissions. The article is well written and can be published in ACP with minor revisions (see below).**

**1. As noted by the authors, the pollutant concentrations depend both on emission rates and atmospheric dilution. To support the author claims it is critical to quantitatively assess the long-term meteorological trends, especially for wind speed (during winter) and nocturnal mixing height (during summer). The sensitivity of pollutant concentration to the meteorological parameters can be studied using a "BC versus wind speed plot" and "BC versus mixing height plot" (the same for NOx and PM2.5).**

We applied the trend analysis to the WS, T and MH, which were shown to be the most important meteorological parameters that affected the PM$_1$ and BC concentration (Teinilä et al., 2019; Järvi et al., 2008). We added time series and the diurnal variation of the MH (Fig. S1 and S2) in the supplementary material. However, we did not include plot of BC vs. meteorological parameters, since we referred to the work by Teinilä et al. (2019) and Järvi et al. (2008), who already studied the sensitivity of the pollutants to different meteorological parameters in the HMA.

The trend analysis of the meteorological parameters did not show any statistically significant changes during the measurement period and therefore we concluded that the variations in meteorology were not the probable cause for the decreasing trends of the pollutants. We omitted the trend analysis of nocturnal MH during summer, since the MH data represented southern Finland and not the city of Helsinki. Therefore, the MH data can not be used to study if, for example, the heat island effect of the growing urban area affected the long-term trends. We added a paragraph in the manuscript explaining the trend analysis of the meteorological parameters:

*One possible cause for the decreased pollution concentrations could have been the changes in the meteorological parameters that affect the dilution. Teinilä et al. (2019) reported that in HMA the two most important meteorological parameter that affected the PM1 concentrations were wind speed (WS) and temperature (T); Järvi et al. (2008) observed that of the meteorological parameters the WS and mixing height (MH) affected the BC concentration the most. The highest concentrations*
5 *were observed at low WS and MH conditions and when the T was either very high in summer or very low in winter, which indicates stable and stagnant meteorological conditions. Also, a temperature decrease during colder periods could increase the emissions from residential wood combustion. Therefore, in addition to BC, we ran the trend analysis for the time series of WS, T, and MH (time series in Fig. S1). However, we did not observe statistically significant trends for any of these parameters. We also studied the trends for the different seasons separately to see, for example, if the temperatures had increased in the*
10 *summer months or decreased in winter months, but this analysis did not yield statistically significant trends either. Therefore, it is likely that the decreasing trends of the eBC concentration were not be explained by the meteorological factors.*

**2. Authors claim that the detached housing areas are influenced by local wood-burning emissions, the assessment being supported by lower NOx/BC ratio at DH sites compared to TR sites. For the sites where Aethalometer was used (DH4**
15 **and DH5) the source contributions should be quantitatively assessed using wavelength dependence of the aerosol absorption (Sandradewi et al., 2008).**

We added a citation to Helin et al. (2018), who applied the model by Sandradewi et al. (2008) to AE33 data measured at DH3 and DH4 (note that the stations are named differently at our study than at the study by Helin et al., 2018). We added a paragraph, which discusses the results by Helin et al. (2018):

20 *The effect of wood combustion at DH sites was studied by Helin et al. (2018) who applied AE33 data measured at TR2, DH3, and DH4 in a source apportionment model suggested by Sandradewi et al. (2008). They reported that on average about 41 and 46 % of the eBC observed at the DH4 and DH5, respectively, originated from wood combustion. The fractions were notably higher than observed at the TR2 (about 15 %). They also observed higher eBC fractions from wood combustion in the cool season: for example, eBC fractions from wood combustion were 46 and 35 % at DH3 in winter and summer, respectively.*

25 *The effect of wood combustion in evenings was also evident in the data by Helin et al. (2018), who observed that both eBC from traffic and wood combustion increased towards the evening at the DH3. A comparison between weekdays and weekends at DH3 showed similar eBC concentrations originating from traffic, but slightly increased eBC concentrations from wood combustion during the weekend.*

30 **3. When comparing BC concentrations measured by MAAP at 637 nm and Aethalometer at 880 nm the wavelength dependence of the aerosol absorption should be taken into account (alternatively all measurements can be reported at the same wavelength). For the reader's convenience, please specify the measurement wavelength of the MAAP on Page 6 Line 12.**

The MAAP wavelength was added in the paragraph about MAAP measurements. As the Aethalometer was compared against the MAAP at 880 nm, we decided to stay with that wavelength. However, we added some discussion about the possible effects of the different wavelengths used. Discussion was added in the end of Sect. 3.1:

*At DH4, DH5, and RB2 at least part of the measurements were conducted by an Aethalometer, which measured the eBC at*
5 *880 nm, which is longer wavelength than on what MAAP operates at (637 nm). This could have caused some difference in the measured eBC concentration in the presence of so-called brown carbon. Brown carbon is organic material, which absorbs light especially at low wavelengths (Andreae and Gelencsér, 2006). However, since the organic carbon absorbs light mainly at wavelengths below 600 nm (Kirchstetter et al., 2004) the difference between the MAAP and Aethalometer wavelengths should not cause a notable effect on the observed eBC concentration.*

---

## Author Response (AR2)

**Author's response to review**
Manuscript: acp-2020-201

We have now corrected all the mistakes noted by the editor. Note, that we also moved one paragraph about the operators of the measurement sites from Sect. 2.2 to Sect. 2.1 where it fit better.

Since all the comments were about mistakes in language or technical issues with references, we do not give a point-by-point reply here. All the changes are visible with red markings in the attached marked up version of the manuscript. Comments regarding the supplement were also fixed.

Sorry for the inconvenience,

Krista Luoma

[revised manuscript text omitted]

---

## Author Response (AR3)

**Author's response to review**
Manuscript: acp-2020-201

Dear Editor,

5   We have now corrected all the typos that were still in the manuscript.

While working with other study involving RB2 data, the main author noticed that part of the eBC data from RB2 were omitted for incorrect reasons. This wrongly omitted data was added in this study and the effect can be seen in Fig. 4d (new and old versions are found next to each other in the attached marked-up manuscript). The time series at RB2 is now more continuous.

10   The $NO_x$ and $PM_{2.5}$ data from the same time periods were also omitted in the older version, but were added to this newer version of the manuscript. The modification of the data sets had only a minor effect on the results (i.e., the values presented in Figs. 2 – 4, S7, and S11 – S13, and in Tables 1 and 3). The most notable effect was in the eBC and $NO_x$ trends observed at RB2 that changed from -7.6 to -6.3 % $yr^{-1}$ and from -4.0 to -4.9 % $yr^{-1}$, respectively. Regarding these changes, a sentence, which now became insignificant, was removed (rows 387 – 390 in the attached marked-up manuscript). **However, the minor**

15   **changes in the values did not change the main results or conclusions made in the manuscript.** I am very sorry for this change at this late state of the manuscript, but I found it important to upgrade the time series, since having a longer and more continuous time series from RB2 makes the observations and results more reliable.

Since all the comments were typos, we do not give a point-by-point reply here. All the changes are visible with red markings

20   in the attached marked up version of the manuscript. One typo found in  the supplement was also fixed.

Sorry for the inconvenience,
Krista Luoma

[revised manuscript text omitted]